# The Influence of the Competences of the Professionals in Charge of Family Evidence-Based Programmes on Internalizing and Externalizing Symptoms in Adolescents

**DOI:** 10.3390/ijerph18052639

**Published:** 2021-03-05

**Authors:** Carmen Orte, Lidia Sánchez-Prieto, Juan José Montaño, Belén Pascual

**Affiliations:** 1Department of Education and Didactics, University of the Balearic Islands, 07122 Palma, Spain; carmen.orte@uib.es (C.O.); belen.pascual@uib.es (B.P.); 2Behavioural Science Methodology, University of the Balearic Islands, 07122 Palma, Spain; juanjo.montano@uib.es

**Keywords:** trainer competences, internalizing symptoms, externalizing symptoms, family-based programmes, the influence of trainers

## Abstract

This study analyses the influence of trainers’ intrapersonal and group management competences on the effectiveness of the Universal Strengthening Families Program 11-14 (SFP 11-14). More specifically, it assesses the effect of these competences on internalizing and externalizing symptoms in adolescents. The analysed data is made up of ratings given by the 174 mothers participating in SFP 11-14. The results confirm the effectiveness of SFP 11-14 in reducing internalizing and externalizing symptoms in adolescents. Using linear regression models, evidence is provided of the influence of the trainers’ expertise, in terms of their competences, in improving internalizing symptoms in adolescents (through a reduction in levels of anxiety, depression, and somatization and in the global internalization scale). Emphasis is placed on how trainer competences can impact on the effectiveness of evidence-based programmes, stressing that this should be taken into account by the public authorities and other stakeholders in the assessment and design of family evidence-based programmes.

## 1. Introduction

### The Role of the Trainer in Family Evidence-Based Programmes: The Universal Strengthening Families Program 11-14

In recent years, there has been a boom in family Evidence-Based Programmes (EBPs), given their good guarantees of effective outcomes [1,2,3,4]. More specifically, family prevention programmes targeted at young people have been promoted [5,6]. This is because parents act as the first agents of socialization for these groups; hence they can exert a key influence at early ages [7,8,9]. So important can this influence be that, with the right parenting models, possible disorders can be averted, such as problems caused by substance abuse or behavioural disorders [6,7,9]. Family-based programmes seek to improve parenting and to give young people the necessary skills to reduce the risk factors that might lead to the onset of these disorders [9,10].

The efficacy of EBP can be guaranteed by validating their principles, components, and strategies [3,4]. With this in mind, rigorous criteria are developed so that the strategies that are followed have been demonstrated to work [10]. In the implementation of programmes, fidelity to these components is fundamental, applying them as stipulated without compromising their effectiveness [2,4]. Linked to this, a key role is also played by the trainer in charge of giving the programme, since they are responsible for ensuring that the standardized manual and the components of the EBP are all properly applied [11,12,13]. The importance of a trainer’s fidelity to a programme has been confirmed in different studies [11,13,14,15]. Nonetheless, few evidence-based programmes targeted at families and youths have assessed trainer competences in this type of programme [16], even though and being responsible for the fidelity of a programme’s implementation, trainers can also influence its outcomes through their level of expertise [17]. Since it is common for trainer profiles and competences to be overlooked in this kind of intervention, a possible bias is being ignored that might compromise the programme’s effectiveness [18,19].

From the scanty available literature that appraises the professionals in charge of evidence-based family prevention programmes for youths, the influence of a series of intrapersonal and group management competences has been demonstrated on both the effectiveness of the results and the implementation process. Table 1 shows the family prevention programmes of this kind that were identified. Only programmes with empirical evidence were selected. A total of five studies explored trainers’ intrapersonal competences, while three focused on group management skills.

In the case of the trainers’ intrapersonal skills, differing ones were identified as being important: (a) agreeableness, (b) pleasantness, (c) flexibility, (d) optimism toward change, (c) organizational powers, (f) engagement, (g) creativity, and h) sociability (the development of social skills). The influence of agreeability (or affability) was highlighted as being a particularly relevant intrapersonal competence for trainers, with its impact on group functioning being confirmed in the “PROSPER” programme [20], on adherence rates in the case of the “Power Coping” programme [21,22] and on fidelity of implementation in the “Family Check Up” (FCU) programme [13]. Klimes-Dougan et al. (2009) [12] found extraversion, which is tied in with this last competence, since it encompasses agreeableness and sociability, to be significantly associated with fidelity of implementation in the Skills for Success Programme. Along the same lines, as Sánchez-Prieto et al. (2020) [19] point out, the participants of prevention programmes believe that it is important for trainers to be pleasant, thus contributing to a dynamic learning process. To achieve this, the trainers must also be creative, because pleasant, creative dynamics—based on roleplays, debates, and reflection on issues—have been shown to boost family participation [5,23,24].

Flexibility (openness to an experience) alludes to a trainer’s capacity to be receptive to new experiences (such as the first time an EBP is implemented) and to be able to adapt to any problems that might arise. Whilst Mauricio et al. (2019) [13] and Klimes-Dougan et al. (2009) [12] refer to it as a personality trait that boosts fidelity to a programme, Feinberg et al. (2007) [20] explain that openness to an experience has a negative influence on the functioning of the team of professionals (see Table 1) [12,16,20]. As for optimism toward change, the “Project Towards No Drug Abuse (TND)” [11] assessed this in combination with training for educators. While training boosted their self-efficacy and fidelity levels, a belief in the possibility of change was negatively associated with the educators’ fidelity. In contrast, other studies point to optimism or perceptions of a programme’s usefulness as one of the variables with the power to predict a more successful implementation [25] or better group functioning [19]. In the “Coping Power” programme [22], an assessment was made of cynicism (scepticism) about organizational change. It was demonstrated that, together with work backdrops with low levels of autonomy, cynicism led to less engagement by trainers. This last finding again emphasizes the importance of believing in a programme’s usefulness and its capacity to bring about change in the participants. 

Using experts and trainers, Sánchez-Prieto et al. (2020) [19] conduct an analysis of trainer profiles and competences. Both the experts and the trainers agree that, in addition to some of the competences mentioned above, trainers must have the capacity to organize the sessions in advance. This is tied in with their capacity for engagement with the criteria of the EBP, the teaching of its contents and the application of its methodologies [18,26].

As for group management competences, the “Coping Power” programme [21] made an important contribution by demonstrating that good group management by trainers led to a lower rise in disruptive behaviours among the adolescents taking part (see Table 1). Another relevant factor was confirmed by Sale et al. (2008) [27]: the influence of the bond between the participants and the trainer on the programme’s efficacy. More specifically, they concluded that more confidence in the trainer and more empathy on the latter’s part led to a significant improvement in the adolescents’ social skills. According to Mazzucchelli and Sanders (2010) [16], by creating an alliance between the trainer and the participants through empathy on the former’s part, the rigidity implicit in EBP can be overcome. The supplied evidence points to the need for further research into the impact of group management and empathy on the effectiveness of programmes. Lastly, the capacity to reinforce change was highlighted by experts and trainers as a key factor in boosting family adherence and motivation [18,19].

Nonetheless, studies have also been identified with no significant trainer-related findings. This is the case of Hodge et al. (2017) [15], who were not able to identify significant predictive variables associated with trainers that would improve the implementation of the “PPP” programme. Despite this, as the authors point out, research into the influence of trainer competences on programme outcomes is limited and when the issue is tackled, studies tend to focus on the trainers’ fidelity in implementing a programme, without taking into account the fact that their competences might influence the end result [28]. Although several authors have recommended research of this kind and a selection process for trainers, few studies assess intrapersonal and group management competences (both of which are important in the implementation of evidence-based prevention programmes) [5,15,22,24,26].

Based on the above premises, the Universal Strengthening Families Program 11-14 (SFP 11-14) assesses the competences of the professionals that implement it, using ratings awarded by the mothers taking part in the programme. SFP 11-14 is an evidence-based family prevention programme aimed at adolescents. Its main aim is to prevent internalizing and externalizing symptoms, boosting protective factors. As a result, it acts on two types of variables: (a) family variables; that is, inappropriate parental upbringing models and family dynamics; and (b) personal variables. To improve the capacity for parenting, the parents receive training in the following skills and strategies: (a) emotional regulation strategies; (b) communication skills; and (c) behavioural strategies to modify behaviour. As for the children, SFP 11-14 focused on giving them the following types of skills and strategies: (a) emotional regulation strategies; (b) communication skills; and (c) coping skills. The programme is based on 6 sessions, working in parallel groups with the children and parents, and the family as a whole. The skills and strategies complement one another and they are reinforced through practical work at home. SFP 11-14 has already been proven to be effective in aiding parenting and family dynamics through training for parents and families [28].

The main aim of this study is to assess the influence of trainers’ intrapersonal and group management competences on internalizing and externalizing symptoms in adolescents. First, an analysis must be made of the effectiveness of SFP-U 11-14 on internalizing and externalizing symptoms in adolescents.

## 2. Methodology

### 2.1. Participants

SFP 11-14 is aimed at families with adolescents aged between 11 and 14, coinciding with the transition from primary to secondary school. The families are from schools that meet the following inclusion criteria: (1) they had not taken part in prevention programmes during the previous two years; (2) they were state or subsidized schools; and (3) they were schools in the Balearic Islands or in the Castilla y León region (both places with professionals with training in the programme).

The programme began with 16 experimental groups and 17 control groups. For this study, only the sample of mothers in contact with the trainers (the experimental groups) was used. The sample was made up of a total of 174 mothers in charge of rating the trainers and the adolescents, with a mean age of 43.85 (SD = 5.16). Most were from nuclear families; that is, made up of parents and children (84.5% of the participants). A total of 55.7% lived in the Balearic Islands and 44.3% lived in Castilla and León (two of Spain’s self-governing regions). A total of 73.5% lived in urban areas.

As for the adolescents who were evaluated, 59.2% were boys and 40.8% were girls. Adolescents in primary school education predominated (69.5%), with an overall mean age of 11.7 (SD = 0.964). As for the trainers who were assessed, most were women (81.7%), with a mean age of 38.62 (SD = 7.769), and most worked in the social services sector (67.6%).

The inclusion criteria for the mothers were as follows: (1) to attend 80% of the sessions, (2) to have children aged between 11 and 14, and (3) not to suffer from a substance-related disorder (assessed prior to session 1). The sample taking part had a final adherence rate of 86.40%.

### 2.2. Design and Procedure

A quasi-experimental methodology was used, with a pretest/post-test design to assess the programme outcomes. The outcomes for the adolescents were assessed using reports completed by the mothers, given out before the SFP-U 11-14 began (pretest) and at the end of session 6 (post-test). The trainers’ competences were also assessed at the end of session 6 by the mothers. To avoid possible biases in the assessment process, the mothers were kept separate from the trainers and the adolescents while they filled in the questionnaires. Prior to its implementation, the programme was explained to the families at presentational meetings at each of the schools taking part. Families were selected who met the inclusion criteria.

### 2.3. Measures

#### 2.3.1. The Behaviour Assessment System for Children and Adolescents: Parent Rating Scales (PRSs) 

To assess adolescent symptoms, the parent rating scale (PRS) was used, a scale from the behavioural assessment system for children and adolescents (BASC) [29]. A decision was taken to use the Spanish validation of version 3, aimed at adolescents aged between 12 and 18 [30]. Although some children were 11 years old, the questionnaire that best fit in with their developmental stage was version 3.

With the PRS, the children were rated by their mothers using 7 clinical scales with a high internal consistency according to Cronbach’s alpha: (1) depression (α = 0.844), (2) anxiety (α = 0.653), (3) somatization (α = 0.866), (4) aggressiveness (α = 0.829), (5) hyperactivity (α = 0.809), (6) attention problems (α = 0.724), and (7) atypicality (α = 0.694). The global internalization scale was created using the depression, anxiety, and somatization scales. It also had a high internal consistency (α = 0.873). Furthermore, the test–retest reliability results show very high correlations of 0.89 for the children, with mean values of 0.85 for the parents. The scale was made up of 68 items, answered using a Likert scale with the following possible responses: (1) never; (2) sometimes; (3) often; and (4) almost always. It took an average of about 15 minutes to answer.

#### 2.3.2. The Trainer Competence Questionnaire

To assess the trainers’ level of expertise, a questionnaire answered on a Likert scale was used, with a design based on theoretical considerations concerning trainer competences [26] (see Table 1). Its purpose was to determine which competences the mothers identified with the trainer. A decision was taken to use a purpose-designed questionnaire in order to assess whether the trainers of SFP 11-14 had the specific competences recommended in the relevant literature [12,13,14,15,20,21,22,27]. The questionnaire was made up of 11 items with 5 possible answers: (1) Very bad; (2) Bad; (3) Average. (4) Good; and (5) Excellent.

An exploratory factor analysis (principal components method) with Varimax orthogonal rotation was conducted, based on Kaiser normalization of the items rated by the mothers in their evaluation of the trainers. The KMO (Kaiser–Meyer–Olkin) measure of sampling adequacy took a value of 0.899, while Bartlett’s test of sphericity had a Chi-square value of 1133.948 (*p* < 0.001), both indicating the obtainment of a suitable factor model. The factor analysis led to the identification of two main factors (see Table 2) according to the Kaiser criterion (eigenvalues > 1), explaining 68.23% of the total variance. The first factor, associated with “intrapersonal competences”, explained 44.64% of the total variance. The second factor, associated with the trainers’ group management competences, explained 23.58% of the total variance. From a reliability analysis of factor 1, a Cronbach α of 0.912 was obtained, while a value of 0.767 was obtained for factor 2. Cronbach α values greater than 0.70 indicate a good internal consistency [31].

### 2.4. Data Analysis

The results were analysed in two steps:(a)A comparison of the pretest and post-test means of the rated internalizing and externalizing symptoms, using the Student’s *t*-test.(b)Linear regression models of the influence of the trainers’ competences on the adolescents’ symptoms. For the selection of the models, an automatic backward elimination process was used for the variables. The factors that were obtained in the factor analysis acted as predictive variables. As response variables, the difference between the pretest and post-test ratings was used, with positive differences indicating an improvement in symptoms after the intervention.

## 3. Results

### 3.1. The Pre- and Post-Test Comparison of Adolescent Symptoms

Table 3 shows the results of the pre- and post-test comparison of the adolescents’ internalizing and externalizing symptoms. More specifically, it shows the mean (M) and standard deviation (SD) at each stage (pretest/post-test), the value of the Student’s *t*-test for the comparison of means, its level of significance (*p*), and the 95% confidence interval for the difference between means.

From the results of the compared pretest and post-test means of the rated symptoms, significant differences were found for all the variables. The adolescents experienced a significant drop in variables associated with externalizing symptoms. More specifically, significant differences were found in levels of aggression (t (165) = 5.182; *p* < 0.001), hyperactivity ((t (165) = 3.853; *p* < 0.001), attention problems (t (165) = 10.914; *p* < 0.001), and atypicity (t (165) = 13.141; *p* < 0.001). 

Likewise, there was also a drop in the variables associated with internalizing symptoms, with significant improvements in levels of anxiety (t (165) = 4.793; *p* < 0.001), depression (t (165) = 5.578; *p* < 0.001) and somatization (t (165) = 6.014; *p* < 0.001) when the pretest and post-test ratings were compared (see Table 3). The most significant differences were related to the global internalization scale (made up of the anxiety, depression, and somatization variables), where a t (165) = 7.217 was obtained that was found to be significant (*p* < 0.001).

As a result, the overall effectiveness of SFP 11-14 was confirmed for all the analysed symptom-related variables.

### 3.2. Linear Regression Models of the Influence of Trainer Competences on Adolescent Symptoms 

Table 4 shows the four selected linear regression models. For each competence factor, it contains the following information on the influence of trainer competences on the adolescents’ symptoms: the coefficient (B), its standard error (SE), the value of the test (t), its significance (p), and the 95% confidence interval for B.

The results of the linear regression show that the trainers’ level of expertise has a significant impact on improvements in the adolescents’ internalizing symptoms. However, factors 1 and 2 were not found to exert any influence on the adolescents’ externalizing symptoms.

In the case of the internalizing symptoms, factor 1, which is associated with the trainers’ intrapersonal skills, was related to improvements in levels of depression and somatization and with the global internalization scale. Hence, the higher the trainers’ intrapersonal skills, the lower the level of depression observed in the adolescents at the end of the programme (B = 1.766; *p* < 0.05). The same applies to the level of somatization (B = 1.793; *p* < 0.01) and to improvements in the global internalization scale (B = 1.783; *p* < 0.05). The F statistic of the ANOVA of the regression model took a value of (F (2, 150) = 5.410; *p* < 0.05).

As for group management competences (factor 2), the higher the trainer’s expertise in these skills, the lower the observed level of anxiety (B = 1.323; *p* < 0.05). Furthermore, better group management skills were associated with an improvement in the global internalization scale (B = 1.233), even though this improvement was not statistically significant (*p* = 0.057).

None of the linear regression models for the variables relating to the externalizing symptoms was significant, and so the trainers’ competences were not found to influence the adolescents’ externalizing symptoms.

## 4. Discussion

Family evidence-based programmes for families and youths have focused on developing strategies and components that, in combination, have proven to be effective in dealing with psychological disorders [4,9,18,32]. This is the case of SFP 11-14, whose multicomponent structure and 6-session version are effective in reducing internalizing and externalizing symptoms in adolescents, showing that it is possible to reduce these symptoms in universal populations through short-length programmes. The results coincide with systematic reviews [6,10] and with the results of other EBP [33,34,35] where families are used as a key strategy in the prevention or reduction of symptoms in adolescents.

Nonetheless, to make sure that EBP are properly implemented, it is fundamental to take into account the trainers’ skills and abilities [18,23,24]. A poorly implemented EBP can comprise the results [3,4,5,36]. That is why current studies assess fidelity of implementation by trainers [11,13,14,17]. However, this paper highlights the fact that as well as assessing fidelity of implementation, it is important to take into account how a trainer’s competences might impact on a programme’s effectiveness. More particularly, it focuses on trainers’ intrapersonal and group management competences, showing that the trainers’ intrapersonal skills lead to improvements in depression and somatization levels among adolescents and in the global internalization scale. Klimes-Dougan et al. (2009) [12] suggested that better outcomes in programmes might be linked to intrapersonal skills, because the trainers are more engaged and cope better with the challenges of the programme. Mauricio et al. (2019) [13] upheld these same ideas, arguing that competences like openness, engagement, and agreeableness contribute to the effectiveness of the trainer’s work. Nevertheless, although the relevance of intrapersonal competences in trainers has been emphasized in literature, few programmes have assessed the influence of these competences on the effectiveness of their outcomes [14,20,22].

Similarly, group management competences lead to improvements in adolescents’ anxiety levels. Lochman et al. (2017) [21] and Sale et al. (2008) [27] also identified a link between group management skills and disruptive behaviour and social skills, respectively. Likewise, other studies have also highlighted the importance of coordination and group management in improving the outcomes of EBP and in effective conflict resolution [37,38]. Along the same lines, Mihalic et al. (2008) [39] found that the use of management techniques by professionals, as opposed to managerial styles, leads to a bigger drop in disruptive behaviours among adolescents. Indeed, experts and academics have coincided in the need for the professionals in charge of evidence-based family programmes to be skilled at group management, communicating, and empathizing [18,26]. The importance of these skills lies in their potential for motivating the participants [12], while also allowing trainers to adapt to the group’s requirements [1] and to create a feeling of confidence and security [14,24]. It is also important to explore the contribution of group management competences to the creation of an alliance between the trainer and the participants [40].

The figure of the trainer is clearly acquiring increasing relevance in the development of EBP directed at families and youths. Given trainers’ potential influence on both the implementation of the programme and its outcomes, they are yet another factor to take into account in the design of EBP. Hence, some authors have already highlighted the importance of assessing their competences and attitudes to EBP prior to the commencement of an implementation [40]. Through an initial assessment, it would be possible to ensure that the trainers have the necessary skills to implement the programme [41]. Indeed, authors like Forehand et al. (2010) [5] recommend a trainer selection process to make sure that the trainer meets the specific requirements of EBP. With the same aim in mind, other authors have pointed to the need for training, highlighting competences to be fostered like social skills or group management [1,23,26]. Authors, like Parrish and Rubin (2011) [42], even propose continuous training as the best option so that the trainers’ work can be supervised and guided. Lastly, it is also advisable for the trainers themselves to be aware of the importance of their own performance and practices in the implementation of programmes, hence boosting their engagement in them [43]. This awareness of their relevance must be conveyed to them during the training process. 

The main contribution of this study lies in the fact that the results have reinforced the need to take trainer competences into account, not just in fidelity of implementation but also in assessments of the effectiveness of programmes. The study was also based on a review of specialist literature, identifying those EBP that take the figure of the trainer into account. To date, few EBP directed at families and youths have assessed the influence of programme implementers on adolescent symptomatology [19,43]. This research study therefore has useful implications on the development of family-based EBP.

It is worth noting that trainer assessments in this study were based on external ratings (by the mothers) to avoid the problem of social desirability that might have occurred if the trainers had assessed the process. The study’s main limitation, on the other hand, concerns the trainer assessment questionnaire. Although the instrument that was used is theoretically grounded, with good psychometric properties, there is no yardstick for comparisons with the population and neither has it been validated. A purpose-designed questionnaire was chosen in order to assess whether the trainers who implemented SFP 11-14 had the specific skills recommended in the relevant literature. 

In future research, a control group of trainers must be formed to establish causal links between professional competences and adolescent symptoms. Likewise, a more in-depth study could be made of the relationship between both factors in order to identify whether specific trainer competences (empathy, creativity, sociability, flexibility, etc.) influence adolescent symptomatology.

## 5. Conclusions

This paper demonstrated the effectiveness of SFP 11-14 in reducing internalizing and externalizing symptoms in adolescents. Furthermore, it has highlighted the influence of trainers’ intrapersonal and group management competences on internalizing symptoms in adolescents. More specifically, intrapersonal competences were observed to have a significant impact on levels of depression and somatization and on the global internalization scale, while group management competences were found to have a significant effect on anxiety levels. Hence, trainer competences have been demonstrated to influence the outcomes of SFP 11-14, highlighting the importance of the figure of the trainer and their competence levels.

Thus, the figure of the trainer must be taken into account by the public authorities and other stakeholders in the evaluation and design of evidence-based programmes aimed at families and youths, given the trainers’ demonstrated influence on programme outcomes. Standard guidelines should be developed for assessing trainers so that they have the necessary competences to guarantee well-implemented EBP [5,23,40].

## Figures and Tables

**Table 1 ijerph-18-02639-t001:** Description of studies of evidence-based prevention programmes (EBP) for youths where the influence of the trainer is assessed.

Authors	Programme	Type of Prevention	Sample	Objective of EBP	Trainer Component	Trainer-Related Outcomes
Intrapersonal competences						
Feinberg et al. (2008) [20]	PROSPER	Universal	159 trainers	Evaluation of the impact of intrapersonal competences (personality traits) on group functioning.	1. Openness to the experience 2. Agreeableness 3. Conscientiousness	Openness to the experience had a negative influence on team functioning. Conscientiousness had positive implications on team functioning.
Eames et al. (2009) [14]	Incredible Years PT Programme (IY-PT).	Selective	86 parents	Evaluation of whether intrapersonal competences (the capacity for observation) can predict changes in parenting skills.	1. Observational powers 2. Fidelity to the programme guidelines	Trainers with good observational powers and fidelity to the programme boost its effectiveness (better parenting skills).
Klimes-Dougan et al. (2009) [12]	Early Risers intervention (Skills for Success)	Selective	27 schools	Evaluation of the impact of intrapersonal competences on fidelity of implementation.	1. Personality traits 2. Expectations and beliefs about the programme’s usefulness 3. Coping with adversities	Traits indicative of high extraversion and low neuroticism, belief in the programme’s usefulness, and good coping skills when faced with adversity are associated with fidelity to the programme.
Lochman et al. (2009) [22]	Coping Power	Indicated	32 trainers	Analysis of the influence of intrapersonal competences on the dissemination of the process (programme delivery and commitment).	1. Agreeableness 2. Conscientiousness 3. Cynicism (versus optimism) about organizational change	Agreeableness on the part of trainers is associated with better adherence by parents. Conscientiousness is associated with greater engagement by the children. Cynicism (scepticism) about organizational change and low levels of autonomy are associated with less engagement by trainers.
Mauricio et al. (2019) [13]	Family Check Up (FCU)	Selective	112 trainers	Evaluation of the impact of intrapersonal competences on fidelity of implementation.	1. Agreeableness 2. Conscientiousness 3. Openness 4. Extraversion 5. Attitudes to evidence 6. Commitment 7. Wellbeing	Professionals with better intrapersonal skills and attitudes to evidence, a stronger sense of commitment and better sense of wellbeing are associated with higher fidelity.
Family management competences						
Sale et al. (2008) [27]	Youth Mentoring Initiative (CSAP)	Universal	100 youths	Evaluation of the influence of family management skills on the prevention of drug consumption (programme efficacy).	1. The capacity for links to be forged between the trainer and the participants (cooperation, self-control, assertiveness, empathy).	Better perceptions of confidence, mutual support and empathy between the participants and the trainers are linked to improvements in social skills.
Hodge et al. 2017 [15]	Triple P–Positive Parenting Program (PPP)	Selective	59 trainers	Evaluation of intrapersonal skills and the implementation of the programme.	1. The influence of peer trainer support on implementations.	None of the variables turned out to be a significant predictor for implementations.
Lochman et al. (2017) [21]	Coping Power	Indicated	180 adolescents	Evaluation of the influence of family management skills on child behaviour.	1. Group management 2. Clinical skills (non-coercive behavioural styles)	A lower rise in disruptive behaviours and behavioural problems in adolescents.

**Table 2 ijerph-18-02639-t002:** Factor loadings of the items for each of the obtained components.

Items from the Questionnaire on Trainer Competences	Components	
	1	2
Factor 1. Intrapersonal competences		
Organized	**0.849**	0.131
Flexible	**0.837**	
Pleasant	**0.834**	0.275
Creative	**0.778**	0.231
Sociable	**0.710**	0.340
Engaged	**0.705**	0.355
Agreeable	**0.676**	0.494
Optimistic	**0.605**	0.430
Factor 2. Group management competences		
Group management skills		**0.892**
Empathic	0.347	**0.797**
Capacity to reinforce change	0.494	**0.589**

Bold items determine components.

**Table 3 ijerph-18-02639-t003:** Pretest and post-test comparison. Difference in means following the implementation of SFP 11-14.

	Pre-Test		Post-Test				95% Confidence Interval for the Difference
	M	SD	M	SD	t (165)	*p*	
Externalizing variables							
Aggression	51.488	10.610	48.734	10.819	5.182	0.001	[1.704–3.801]
Hyperactivity	54.818	11.651	52.345	11.674	3.853	0.001	[1.205–3.739]
Attention problems	52.849	11.127	50.054	10.914	4.747	0.001	[1.63–3.957]
Atypicity	52.500	14.425	48.825	13.141	5.471	0.001	[2.34–5.000]
Internalizing variables							
Depression	53.204	13.787	49.861	11.850	4.793	0.001	[1.965–4.720]
Anxiety	50.885	11.379	47.265	11.014	5.578	0.001	[2.338–4.902]
Somatization	50.824	11.506	47.193	11.274	6.014	0.001	[2.438–4.822]
Global internalization							
Scale	52.539	12.956	48.060	12.307	7.217	0.001	[3.253–5.704]

**Table 4 ijerph-18-02639-t004:** Selected linear regression models: the influence of trainer competences on adolescent symptoms.

Variable	B	SE	t	*p*	95% Confidence Interval for B
Depression					
Factor 1: Intrapersonal competences	1.766	0.788	2.242	0.026	[0.210–3.322]
Anxiety					
Factor 2: Group management competences	1.323	0.659	2.008	0.046	[0.021–2.625]
Somatization					
Factor 1: Intrapersonal competences	1.793	0.646	2.774	0.006	[0.516–3.071]
Internalization					
Factor 1: Intrapersonal competences	1.783	0.688	2.592	0.010	[0.0424–3.142]
Factor 2: Group management competences	1.233	0.642	1.922	0.057	[−0.035–2.501]

## Data Availability

Data available on request due to privacy/ethical restrictions.

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
