# Peer review of "The Influence of the Competences of the Professionals in Charge of Family Evidence-Based Programmes on Internalizing and Externalizing Symptoms in Adolescents"

_ijerph, 2021, doi:10.3390/ijerph18052639_

Round 1
Reviewer 1 Report
The english language and style are fine just requiring minor spell check (e.g., in the Discussion section, in the second paragraph, it is written: "A poorly implemented EBP can comprise the results", instead of can compromise. Futhermore, it is not clear the procedures that led to the construction of the trainer competence questionnaire (the author of the questionnaire is also not revealed).
1. Areas of strength
a. Relevance of the topic
b. Exhaustive and current revision of the literature concerning trainer-related outcomes
c. The results have useful implications for the development of evidence-based family programs
2. Areas of weakness
a. Use of a quasi-experimental methodology
b. Reduced number of investigations (a total of 5 studies) focusing on trainer related outcomes
c. The assessment of the trainers' skills was carried out only by the mothers
d. The theoretical foundations that led to the construction of the instrument used to assess the trainer's skills are not clearly stated
Author Response
Dear Referee,
Many thanks for all your comments and insights into the paper. They have been very helpful in improving our proposed article. Please find below the responses to each of your comments. Following your instructions, we have made the following changes:
- The spelling errors pointed out by the reviewer have been corrected.
- As we have also explained to Reviewer 2, all the articles on which the design of the trainer assessment questionnaire was based have been referenced, hence defining the theoretical foundations on which it was built.
- The limitations of the trainer assessment questionnaire have been included in the discussions section.
- The conclusions and discussions sections have been improved upon, based on the weak points indicated by the reviewer.
We hope that our replies are sufficient. We will gladly provide any further explanations and comments that might be needed.

Reviewer 2 Report
It is a good paper on how the competences of professionals influence the effectiveness of evidence-based programmes. Good theoretical and methodological planning. Perhaps, I would have liked to delve deeper into the discussion of the results and for the conclusions to advance lines of work with the professionals, and know the limitations of the study
Author Response
Dear Referee,
We would like to thank you for your review and for the guidance that you have provided. We have followed your recommendations and believe that they have improved on the previous version. We hope that our brief replies to your comments are sufficient.
Specifically, the following modifications have been made to the article:
- The discussions section has been expanded, including reflections on lines of work with professionals. A description of the main limitations of the study has also been incorporated.
2. The article mentioned that one of the main limitations was the trainer assessment questionnaire. Because it is purpose-designed, there is no yardstick for comparisons with the population, and neither has it been validated. Nonetheless, it is theoretically grounded and it was found to have good psychometric properties. Despite its constraints, a decision was taken to use this questionnaire in order to assess whether the trainers of SFP 11-14 had the specific competences recommended in the relevant literature. This comment has been moved from the conclusions to the discussions section.
We hope that our replies are sufficient. We will gladly provide any further explanations and comments that might be needed.

Reviewer 3 Report
The purpose of the study was to examine trainer characteristics on internalizing and externalizing symptoms in adolescents. This interesting study. I have a few comments below.
Introduction: I think it would be helpful if you could provide a little more information about the Strengthening Families Program for the reader. You might want to mention goals of the program and the topics covered for both parents and children through the use of group leader manuals and handouts etc.
Methods: I am curious as to why the authors decided to use the version of the BASC for 12-18 year-old children when the participants in the program were between 11-14? You might want to include a justification for this.
Results: The factor loadings for optimism and capacity to reinforce change were a little low. It would be interesting to see which of the intrapersonal competencies or group management competencies were important for depression, anxiety, somatization and internalization. The authors might consider making these analyses available in an online appendix if this is an option.
Discussion: I did not see a discussion of strengths and weakness of this study. Usually this is included in the discussion.
Author Response
Dear Referee,
We appreciate your review of the study and your comments, which have helped to improve the quality of the article. We hope that our replies to your remarks are sufficient. We will gladly provide any further explanations and comments if necessary.
In reference to your recommendations and other observations, the following modifications have been made to the article:
- As Reviewer 2 indicates, we believe that a description of SFP 11-14 is needed, so an outline has been included of the following characteristics of the programme: objectives, strategies and skills, structure and functions.
- A justification of the use of BASC has been included, as requested. Version 3 of BASC was used because although some children were 11 years old, this version of the questionnaire best fits in with their developmental stage. In addition, most of the children were between 12 and 14 years old, corresponding to BASC version 3.
- We appreciate your recommendation that a specific causal relationship should be established between trainer competences and adolescent symptoms. We have incorporated it into the discussions section as a possible line of research.
- The discussions section has been reinforced, incorporating the strengths and limitations of the study and future lines of research.
We hope that our replies are sufficient. We will gladly provide any further explanations and comments that might be needed
